# Medusa: Unveil Memory Exhaustion DoS Vulnerabilities in Protocol Implementations

## ABSTRACT

Web services have brought great convenience to our daily lives. Meanwhile, they are vulnerable to Denial-of-Service (DoS) attacks. DoS attacks launched via vulnerabilities in the services can cause great harm. The vulnerabilities in protocol implementations are especially important because they are the keystones of web services. One vulnerable protocol implementation can affect all the web services built on top of it. Compared to the vulnerabilities that cause the target service to crash, resource exhaustion vulnerabilities are equally if not more important. This is because such vulnerabilities can deplete the system resources, leading to the unavailability of not only the vulnerable service but also other services running on the same machine. Despite the significance of this type of vulnerability, there has been limited research in this area.

In this paper, we propose Medusa, a dynamic analysis framework to detect memory exhaustion vulnerabilities in protocol implementations, which are the most common type of resource exhaustion vulnerabilities. Medusa works in two phases: exploration phase and verification. In the exploration phase, a protocol property graph (PPG) is constructed to embed the states with relevant properties including memory consumption information. In the verification phase, the PPG is used to simulate DoS attacks to verify the vulnerabilities. We implemented Medusa and evaluated its performance on 21 implementations of five protocols. The results demonstrate that Medusa outperforms the state-of-the-art techniques by discovering overall 127× maximum memory consumption. Lastly, Medusa has discovered six 0-day vulnerabilities in six protocol implementations for three protocols. Particularly, one of the vulnerabilities was found in Eclipse Mosquitto, which can affect thousands of services and it has been assigned with a CVE ID.

**ACM Reference Format:**
Anonymous Author(s). 2023. Medusa: Unveil Memory Exhaustion DoS Vulnerabilities in Protocol Implementations. In *Proceedings of ACM Conference (Conference'17)*. ACM, New York, NY, USA, 11 pages. https://doi.org/10.1145/nnnnnnn.nnnnnnn

## 1 INTRODUCTION

Denial-of-Service (DoS) attacks have emerged as a prevalent form of attacks against web services in the past two decades [48]. According to [26], DoS attacks can take two forms. The first form targets to overwhelm the network bandwidth of the target service

with a massive amount of useless traffic, while the second form exploits vulnerabilities present in the target service. The majority of existing research has focused on detecting the first form of attack by monitoring incoming traffic [3, 21, 25, 31, 36, 40]. In contrast, less attention has been paid to DoS attacks through vulnerabilities.

Although less research effort was devoted to DoS vulnerabilities, they are important subjects for studying. The reason is that attackers can cause the same or even greater harm with less effort by exploiting vulnerabilities than by overwhelming a service with massive traffic. Moreover, DoS vulnerabilities inside protocol implementations are of greater concern. This is because every service relies on certain protocols to communicate and protocol implementations are the necessary building blocks of web services. Vulnerabilities in one protocol implementation can affect multiple services, amplifying their impact. Therefore, we focus on studying DoS vulnerabilities in protocol implementations [1].

DoS vulnerabilities can be classified into two types [44]: the first type crashes the target service (e.g., due to memory errors) while the second type exhausts the resource of the host machine (e.g., due to excessive memory consumption). Detecting the crashing type of vulnerability has been addressed in established research effort [12, 30, 37–39], while the resource-exhaustion type of vulnerability has received less attention. According to our study on all the protocol-related resource exhaustion DoS CVEs from 2015 to 2022 (205 CVEs in total), 132 (64%) CVEs are related to memory exhaustion. Therefore, we focus our research on studying memory exhaustion DoS vulnerabilities.

To mitigate the risk of memory exhaustion vulnerabilities in protocol implementations, early identification of these issues is imperative. However, unlike memory errors, memory exhaustion vulnerabilities do not have distinct code patterns, making them difficult to detect through static analysis techniques. As a result, it is necessary to use dynamic analysis techniques to capture the behavior of excessive memory usage during the execution of the target protocol program.

Detecting memory exhaustion DoS vulnerabilities in protocol implementations using dynamic analysis techniques presents a unique set of challenges. Firstly, in order to identify potential vulnerabilities, it is necessary to explore the memory consumption of different protocol states. This requires a well-planned strategy that properly schedules the exploration of different possibilities. Secondly, excessive memory usage does not always indicate a high risk of DoS attack. A single message may cause a protocol program to consume significant memory, but may not necessarily be used to launch a DoS attack due to various complex factors such as protocol-imposed rate limitations on specific types of messages. As such, further verification is necessary to assess the viability of DoS attacks. Thirdly, protocols can have multiple implementations in

---

[1] In this paper, we use the terms *protocol implementation* and *protocol program* interchangeably.

different programming languages, and memory exhaustion DoS vulnerabilities can potentially exist in all kinds of languages. To ensure generality, programming language-agnostic analysis techniques are required. However, these techniques limit the information that can be utilized.

In this paper, we propose a dynamic analysis framework called MEDSUA [2], to solve the challenges and unveil memory exhaustion DoS vulnerabilities in protocol implementations. MEDUSA comprises two phases: *exploration* and *verification*. The key object connecting the two phases is the *protocol property graph* (PPG) which embeds the protocol states and relevant properties such as memory consumption information. During the exploration phase, MEDUSA uses a state-aware fuzzer to build and refine the PPG. In return, through querying the PPG, the fuzzer can better explore the memory consumption capability of different states. During the verification phase, MEDUSA generates message sequences by querying the PPG with restrictions and validates viable DoS attacks with the message sequences under simulated environments. PPG can provide guidance to the state-aware fuzzer to substantially explore potentially vulnerable states, addressing the first challenge. The PPG-based verification helps to validate potential DoS attacks, addressing the second challenge. The construction and usage of the PPG do not require program instrumentation to get internal information about the protocol implementation, addressing the third challenge.

We implemented MEDUSA and evaluated its performance with extensive experiments. To evaluate the exploration ability of MEDUSA, we conducted 17,640 CPU hours of experiments on 21 implementations of 5 protocols. Compared to the baseline, MEDUSA can discover overall 125.7× maximum memory consumption. To evaluate the verification ability of MEDUSA, we built the experimental environment and conducted simulated DoS attacks. The results show that the DoS attacks with attack inputs generated from MEDUSA caused target protocol programs to consume significantly more memory and result in worse availability. Moreover, during the evaluation, we discovered six 0-day memory exhaustion DoS vulnerabilities, one of which has been assigned with a unique CVE ID.

In summary, we make the following contributions:

- **Empirical Study.** We conducted an empirical study on resource exhaustion vulnerabilities in protocol implementations, which is the first in this field.
- **Protocol Property Graph.** We proposed a protocol property graph (PPG) based strategy to explore the memory consumption of different states and verify the DoS vulnerabilities.
- **MEDUSA Framework.** We implemented MEDUSA as a dynamic analysis framework and will release the source code for future research.
- **Real-world Impact.** We evaluated MEDUSA with extensive experiments and found six 0-day memory exhaustion DoS vulnerabilities in six protocol implementations, which can affect thousands of web services.

This paper is coupled with a website: https://sites.google.com/view/medusa-dos. We will release the raw experiment data and the source code of MEDUSA on this website.

---

[2] Medusa is a character in the Geek mythology who can turn those who gazed into her eyes into stones.

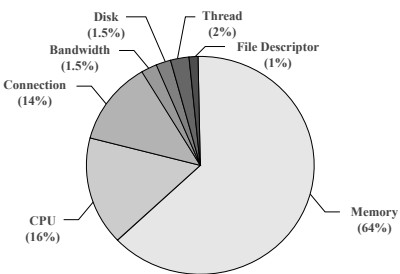

**Figure 1: The distribution of resource exhaustion vulnerability types in protocol implementations from 2015 to 2022**

## 2 BACKGROUND & MOTIVATION

### 2.1 Resource Exhaustion Vulnerability

As defined in [4], resource exhaustion vulnerability refers to a specific fault that causes the consumption or allocation of some resource in an undefined or unnecessary way, or the failure to release it when no longer needed, eventually causing its depletion. In the scenario of protocol implementation, resource exhaustion vulnerability occurs due to improper handling of resource consumption or allocation in specific protocol states.

### 2.2 Empirical Study of Resource Exhaustion Vulnerabilities

To gain a better understanding of resource exhaustion vulnerabilities in protocol implementations, we conducted an empirical study on the Common Vulnerabilities and Exposure (CVE) [13] database which contains a set of publicly disclosed security vulnerabilities. Details of how we collected the data can be found in Appendix A or on our website [2]. In total, we identified 205 memory exhaustion vulnerabilities in protocol implementations from 2015 to 2022. Fig. 1 illustrates the distribution of resource exhaustion vulnerability types. Among the 205 resource exhaustion vulnerabilities, 132 (64%) are related to memory, 32 (16%) are related to CPU, 29 (14%) are related to connections and few are related to other types (disk, thread, file descriptor, and bandwidth). The results indicate that memory is much more vulnerable to resource exhaustion DoS attacks compared to other types, which highlights the significance of studying the detection of memory exhaustion vulnerability.

### 2.3 Motivating Example

We use CVE-2017-7651 and its regression vulnerability discovered by MEDUSA as an example to discuss the motivation behind our proposed technique. CVE-2017-7651 is a memory exhaustion vulnerability in Mosquitto [16], a popular C language implementation of the MQTT protocol which is used by thousands of types of IoT devices. Fig. 2 depicts the patch for CVE-2017-7651 (lines 4-6) and the state transition route to trigger a new memory exhaustion vulnerability. The root cause of the original vulnerability was that the length of the payload requested by unauthenticated users was not checked during the connection phase (starting from line 3) of the MQTT protocol. As shown in the PoC Pseudocode [9], the reporter of CVE-2017-7651 conducted simulated DoS attacks using the CONNECT commands with large payloads, causing Mosquitto program to exhaust its memory and killed by the system. To fix

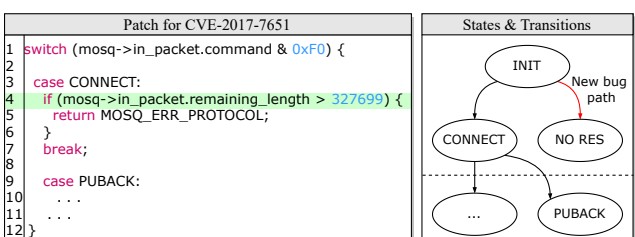

**Figure 2: The patch for CVE-2017-7651 and a new state transition route discovered by Medusa to trigger a new memory exhaustion vulnerability.**

CVE-2017-7651, a length restriction on the CONNECT command was applied (line 4 in the patch). The developers' intuition behind this fixing strategy is that CONNECT command is the only dangerous command that can be manipulated by unauthorized attackers to conduct memory exhaustion DoS attacks. In other words, subsequent states (in the dashed box) can only work after the CONNECT command is passed with authorization. Attackers without compromising the CONNECT command cannot further exploit vulnerabilities in subsequent states.

Despite the previous attempt of fixing the vulnerability, Medusa finds another transition route from initial state to no-response state (as indicated by the red arrow line in Fig. 2) that can result in huge memory consumption. Since this transition does not need to pass through the connect state, we identify it as a new vulnerability. Further analysis revealed that this vulnerability is caused by the wrong order of memory allocations. Specifically, when sending a valid MQTT command excluding CONNECT(such as SUBSCRIBE) with a large payload, Mosquitto will allocate memory for the entire packet before rejecting it, thereby enabling the attacker to exhaust Mosquitto's memory. We promptly reported this vulnerability to the Mosquitto team, who quickly confirmed it and assigned it with CVE-XXX-XXX (the specific number is omitted to maintain anonymity). The Mosquitto team assigned this vulnerability a high (7.5) CVSS score [18] and recognized its DoS threat with the following comment.

> "I confirm that this is a regression. If a malicious client sends as its first command a valid MQTT packet that is not a CONNECT command, Mosquitto will attempt to allocate memory for the entire packet before rejecting it. This means it is possible for a malicious client to cause significant memory use and a denial of service."
>
> —— Mosquitto team

From the example in Fig. 2, we summarize four requirements for detecting memory exhaustion vulnerabilities in protocol implementations. **RequirementI: exploring the memory consumption of different protocol states.** Without awareness of the state transition and related memory consumption, it is impossible to effectively identify the vulnerable memory consumption on the state transition from the initial state to the no-response state, which is crucial in revealing the vulnerability. **RequirementII: using simulated DoS attacks to verify vulnerabilities.** Almost all the cases in our empirical study, including the motivating example, are verified by sending multiple attack packets simultaneously to replicate the real-world attacks. **RequirementIII: generality on various**

**Table 1: Satisfactory status of the four requirements for detecting the memory exhaustion DoS in protocol implementations.**

| | RequirementI (Ability) | RequirementII (Verifiability) | RequirementIII (Generality) | RequirementIV (Optimization) |
|---|---|---|---|---|
| MemLock | ✘̶ | ✘ | ✘ | ✘ |
| AFLNet | ✘̶ | ✘ | ✓ | ✘ |
| Medusa | ✓ | ✓ | ✓ | ✓ |

**programming languages.** Unlike memory corruption vulnerabilities that mainly exist in memory-unsafe languages such as C/C++, memory exhaustion vulnerabilities exist in almost all programming languages. Therefore, a promising detection technique should be program language agnostic. **RequirementIV: optimizing for exploring memory consumption.** To explore memory consumption efficiently, some optimizations (how to select and mutate inputs to explore more memory consumption) need to be taken into consideration. Especially for protocol implementations whose inputs are sequences of messages, we can decide how to select and mutate these messages according to the memory consumption caused by each single message.

Existing dynamic analysis techniques do not satisfy all of the above requirements. The most related techniques to Medusa are algorithmic complexity and protocol fuzzing. Algorithmic complexity fuzzing attempts to find the worst-case resource usage of the entire program, MemLock [43] is a represent tool for testing memory resource usage. Protocol fuzzing aims at fuzzing protocol, AFLNet [38] is a represent tool as it contains all basic main components for fuzzing protocol. As shown in Table 1, both MemLock and AFLNet have limitations in discovering memory exhaustion vulnerabilities in protocol implementations. ❶ MemLock is not state-aware and targets general stateless programs. AFLNet is stateful but does not obtain and explore memory consumption. Without knowledge of the memory consumption of protocol states, we cannot effectively explore the memory consumption of different states and thus miss the opportunity to find the memory exhaustion vulnerabilities in specific states. Therefore, both MemLock and AFLNet **do not fully meet RequirementI.** ❷ Both MemLock and AFLNet rely on the crashing of programs during the fuzzing process to identify vulnerabilities. However, memory exhaustion vulnerabilities may exist even if a single input does not crash the program. Unfortunately, simulating DoS attack for each input in the fuzzing process is not feasible as it introduces tremendous overhead, making MemLock and AFLNet unable to discover these vulnerabilities and **fail to meet RequirementII.** ❸ MemLock needs to instrument programs to gather memory consumption information, but this technique only works for C/C++ programs. This limits its scalability and makes it unable to detect memory exhaustion vulnerabilities in protocol implementations written in other programming languages. Thus, MemLock **fails to meet RequirementIII.** ❹ Both selection and mutation components in MemLock and AFLNet do not optimize for exploring memory consumption and thus **fail to meet RequirementIV.**

These analyses motivate and inspire the design of Medusa to meet all four requirements.

# 3 METHODOLOGY

Medusa introduces *protocol property graph* (PPG) to describe the memory consumption behavior of protocol states. Fig. 3 shows the overview. The overall inputs of Medusa include the protocol implementation for testing, the initial testing seeds, and some miscellaneous information. The overall outputs of Medusa are the PoC message sequences that can trigger memory exhaustion DoS of the target protocol implementation. Medusa works in two phases: exploration and verification. During the exploration stage, Medusa attempts to explore different states of the protocol implementation and measures the memory consumption incurred by each message. Medusa stores the state transition and memory consumption information in the PPG. With the PPG, Medusa can construct sequences of messages and launch simulated attacks to verify potential vulnerabilities in the target protocol implementation.

The exploration phase contains three steps: ❶ Medusa selects a promising message sequence from the pool as the seed. The message sequences are evaluated with the information from the PPG. ❷ Medusa mutates the selected message sequence to create new test inputs. ❸ Medusa feeds the test inputs to the target protocol implementation and monitors for runtime performance such as memory consumption incurred by each message. The verification phase involves two steps: ❹ Medusa builds the attack message sequences according to certain restrictions (such as user-specified states to avoid). ❺ With the attack message sequences, Medusa launches attacks in a configurable simulated environment and reports the viable DoS attack message sequences.

During the exploration phase, The PPG can guide the state-aware fuzzer to explore memory consumption of different protocol states, **satisfying RequirementI**. ❸ can obtain memory consumption and protocol states information and construct PPG without program instrumentation, thus **satisfying RequirementIII**. ❹❺ leverage PPG to simulate DoS attacks and validate potential memory exhaustion vulnerabilities, **satisfying RequirementII**. ❶❷ are guided by PPG to decide how to select and mutate inputs, **satisfying RequirementIV**

## 3.1 Protocol Property Graph (PPG)

Protocol property graph (PPG) is the key concept of Medusa. The definition of *property graph* is as follows:

Definition 1 (PPG). *A PPG is a directed,edge-labeled, attributed graph $G = (V, E, \lambda, \mu)$ with:*

- $V = V_{state} \cup V_{message}$
- $E = E_{sm} \cup E_{ms}$
- $\lambda = \lambda_{sm} \cup \lambda_{ms}$
- $\mu = \mu_{state} \cup \mu_{message}$

*where $V_{state}$ is a set of state nodes, $V_{message}$ is a set of message nodes, $E_{sm}$ is a set of directed edges pointing from a node in $V_{state}$ to a node in $V_{message}$, $E_{ms}$ is a set of directed edges pointing from a node in $V_{message}$ to a node in $V_{state}$, $\lambda_{sm}$ is a set of labeling functions to label edges in $E_{sm}$, $\lambda_{ms}$ is a set of labeling functions to label edges in $E_{ms}$, $\mu_{state}$ is a set of functions to assign properties to nodes in $V_{state}$, and $\mu_{message}$ is a set of functions to assign properties to nodes in $V_{message}$*

Fig. 4 shows an example PPG for the protocol implementation of the motivating example in Fig. 2. The PPG contains all the information used by the exploration phase and required by the verification phase. Medusa can use queries similar to the *Cypher Query Language* [34] to interact with PPG. The queries used by Medusa are made up of the following four clauses:

MATCH  The MATCH clause is the most common clause used by almost every query except for those which create new nodes. It is used for finding a set of nodes and edges matching a given graph pattern in the PPG.

WHERE  The WHERE clause is used to add constraints to the pattern used by MATCH.

CREATE  The CREATE clause is used to create new nodes or edges.

SET  The SET clause is used for creating or updating the properties of nodes and edges.

## 3.2 Exploration

During the exploration phase, Medusa updates the PPG according to the feedback from the Runtime Monitor and reads the information from the PPG for decision-making of the Seed Selector.

### 3.2.1 Runtime Monitor.

The task of the Runtime Monitor is to construct and update the PPG during the execution of each test input. The process is shown in Fig. 5. Every test input is a sequence of request messages, which are sent to the target protocol implementation in order. ❶ First, Medusa starts building the PPG with an initial state (denoted as *S0* in Fig. 5). This is achieved with the following query:

$$\texttt{CREATE (x:State, \{id: S0\})}$$

where x is the variable name, which can be used for further processing (but no use in this query); the parenthesis () indicates that the object is a node; State means that this node is from $V_{state}$; {id:S0} shows the properties of the node. ❷ After that, Medusa sends the first message of the test input to the target protocol implementation and waits for the response. ❸ Upon receiving the response, Medusa will identify the state of the protocol based on the response content and the user-provided specification. The identification of states in Medusa is very similar to how it is done in AFLNet [38]. If a new state is identified, Medusa will use the same query as how it creates *S0* to create a new node for the state. Meanwhile, it will also create a new message node with the following query:

```
CREATE (x:Message, {id: M1, content: ...}, memory: 0KB,
        select_num: 0, update_num: 0, score: 0)
```

where Message indicates that the node belongs to $V_{message}$; content is the raw content (raw bytes) of the message; memory is the memory consumption incurred by the message; memory is the amount of memory consumption incurred by the message and the default unit is KB; select_num is how many times the message has been selected for mutation to generate new test inputs; update_num is how many times the memory property has been updated for this message node; score is used to describe how good the message is to serve as the seed for generating new test inputs, which is calculated based on the previous three properties (Section 3.2.2). Last but not least, Medusa will create the edges connecting the new messages and states. Assume the states are *S0* and *S1*, and the message is *M1*, the query to create the new edges is:

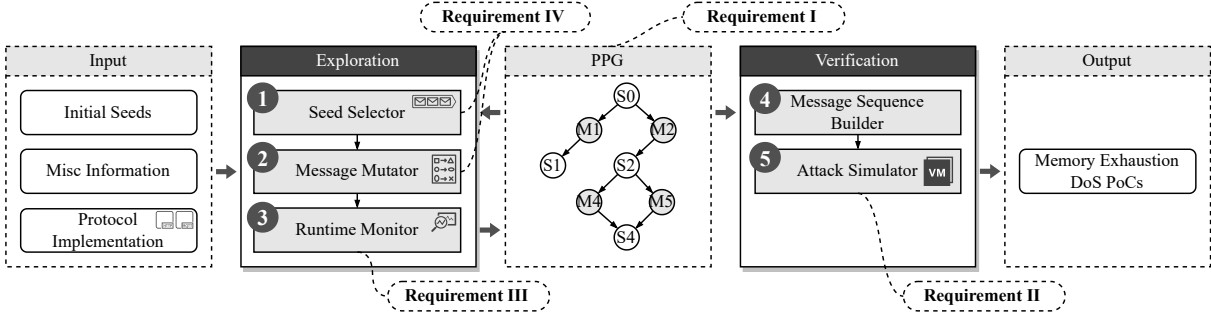

Figure 3: Overview of Medusa

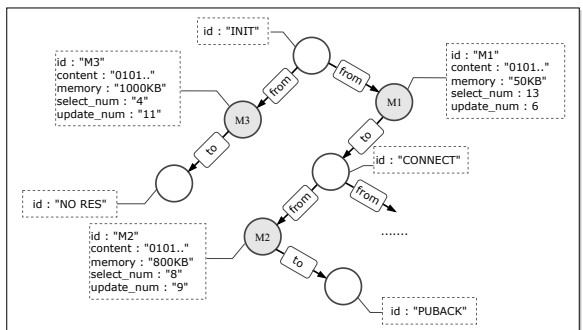

Figure 4: PPG for the motivating example

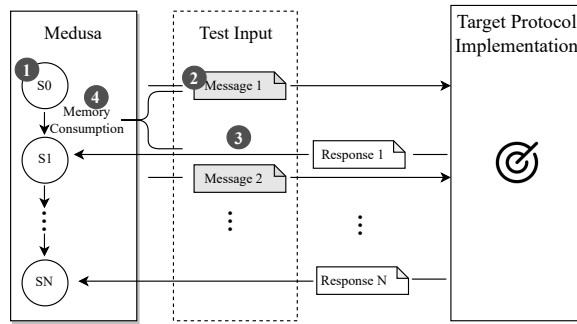

Figure 5: Runtime Monitor workflow

```
           MATCH (s0:State), (s1:State), (m1:Message)
    WHERE s0.id = 'S0' AND s1.id = 'S1' AND m1.id = 'M1'
       CREATE (s0)-[r1:FROM]->(m1), (m1)-[r2:TO]->(s1)
```

where the symbol (a)-[b]->(c) represents an edge *b* pointing from node *a* to node *c*; *r1* belongs to $E_{sm}$ and *r2* belongs to $E_{ms}$. ❹ Apart from creating new nodes and edges, Medusa also checks the memory consumption of the target protocol implementation incurred by the newly sent message. To get the memory consumption information, Medusa accesses Linux's */proc* Filesystem [10] which can monitor resources at the system level without program instrumentation. If the message node is newly created or the message can cause greater memory consumption, Medusa will set the `memory` property of the message. Assume the message has an `id` of *M1* and its incurred memory consumption is 5KB, Medusa will use the following query to update the relevant information:

```
             MATCH (m1:Message) WHERE m1.id = 'M1'
SET m1.memory = '5KB', m1.update_num = m1.update_num + 1
                 , m1.score=Score(m1)
```

where `Score(m1)` is used to calculate the evaluation score of *m1*, which will be explained in Section 3.2.2. Besides, if the message is not a new message and it incurs more memory consumption, Medusa will also update the `content` property to replace the content of the old message with the content of the new one inside the `SET` clause.

### 3.2.2 Seed Selector.
Medusa maintains a pool of seed test inputs and it generates new test inputs by mutating the seed inputs. This helps Medusa to gradually generate better and better test inputs. Given a large number of seed inputs, it is important to decide which inputs should be prioritized. Choosing the messages with high memory consumption

can help to generate test inputs with high memory consumption. However, always selecting messages with high memory consumption can end up in the local optimum. Therefore in Medusa, given a message *m*, three factors are used to evaluate its potential to bring in better test inputs:

(1) Memory consumption $MC(m)$. This factor indicates the max memory consumption on the state transition. It is intuitive that mutating a message with high $MC(m)$ leads to a higher likelihood of discovering larger memory consumption.

(2) Update number $UN(m)$. This factor indicates how many times the memory consumption property of *m* has been updated. The rationale is that if the memory consumption of *m* is updated frequently, mutating *m* is more likely to yield better results.

(3) Select number $SN(m)$. This factor indicates how many times the message has been selected for mutation. If a message has already been selected many times, we should avoid using it too much in order to avoid starving other seeds.

With these three factors, the score for evaluating the potential benefit of mutating a message *m* is calculated with equation 1 as follows:

$$Score(m) = \frac{\ln MC(m) \cdot \ln UN(m)}{SN(m) + 1} \quad (1)$$

Note that as discussed in Section 3.2.1, every time the `memory` property is updated, the `score` property is also updated. Moreover, every time a message is selected for mutation, its `selected_num` property is increased by 1. With the scores calculated, Medusa selects the message with the highest value of the `score` property and use it to generate new test inputs with the following query:

```
            MATCH (m: Message)
      RETURN m ORDER BY m.score DESC LIMIT 1
```

### 3.2.3   Message Mutator.

After selecting the message to mutate, Medusa takes out the corresponding message sequence from the test input pool and locates the exact message. Medusa then mutates on the located message to produce a new test input to execute. Besides the classic mutation operators (e.g. bit flip) used in general fuzzing [30], protocol fuzzing has a special mutation operators type called message-level mutation. Message-level mutation mutate a seed at message granularity, in Medusa, there are three message-level operators: 1) *Replace*. Replace the current message with a message from another seed. 2) *Insert_begin*. Insert a message from another at the begin of the current message. 3) *Insert_end*. Insert a message from another seed at the end of the current message.

All three operators need to choose a message from another seed to replace or insert. Benefit from runtime monitor described in Section 3.2.1, Medusa can get triggered resource consumption of each message in the seeds. The messages which trigger more resource consumption will get higher probability to be chosen by message mutator.

## 3.3   Verification

### 3.3.1   Message Sequence Builder.

During the verification phase, Medusa will first generate message sequences suitable for launching DoS attacks. The generation of the message sequences can be configured according to user requirements. By default, Medusa will prefer the message with the largest memory consumption. It will use the corresponding message node as the starting point, find all the paths to the initial state node on the PPG, and use the messages on whichever path consumes the most memory as the message sequence for launching attacks. Medusa also allows users to add additional properties to the state nodes so that when building the message sequences, the graph traversal can avoid the states with the user-specified properties. The query to rule out the messages on the subgraph of the paths is:

```
                  MATCH
(s1:State)-[r1:FROM]->(m:Message)-[r2:TO]->(s2:State)
      WHERE NOT HAS(s1.user_property) AND NOT
              HAS(s2.user_property)
                 RETURN m
```

where `user_property` is the user-specified property. For example, in the motivating example (Section 2.3), the user can rule out the `Connect` state in this way.

### 3.3.2   Attack Simulator.

Attack simulator validates whether the attack inputs generated from the message builder can cause the DoS of test programs. The simulating process is as following steps: 1) Setting up an experimental DoS environment with the test program running in it. 2) Creating "*attackers*" to send selected attack inputs to the test program. 3) Monitoring the status of the test program under attack. There are several parameters that can be configurable for the attack simulating process: ❶ `system memory limitation`, this parameter decides the maximum memory of the experimental environment. ❷ `monitoring duration`, this parameter decides the total time of conducting the simulation. ❸ `attack time`, this parameter decides

the time to start attacking after the environment has been set up. ❹ `attack intensity`, this parameter decides how many attack inputs sending to the test program every second.

## 4   IMPLEMENTATION & EVALUATION

The implementation of Medusa is comprised of three main parts: a state-aware fuzzer for the exploration phase, a PPG, and an attack simulator for verification. The state-aware fuzzer is built upon AFLNet [38]; PPG is implemented using graphviz [23]; the attack simulator is implemented with Python 3.8.14. Further implementation details can be found in Appendix B or on our website [2]

In the evaluation, we aim to answer the following research questions with experiments:

**RQ1:** How well does Medusa perform in profiling the memory consumption during the exploration phase?
**RQ2:** How well does Medusa perform in simulating DoS attacks during the verification phase?
**RQ3:** Can Medusa discover previously unknown memory exhaustion DoS vulnerabilities in real-world protocol implementations?
**RQ4:** How effective is the optimization of seed selector and message mutator components in the exploration phase? (Appendix E)

## 4.1   Experiment Setup

**Evaluation Baseline** Medusa's fuzzer is built upon AFLNet. We compared Medusa with AFLNet to evaluate its improvement for exploring memory consumption. However, AFLNet is designed for exploring protocol states and does not record memory consumption, we had to make some adjustments to it in order to make it feasible for comparison. Specifically, we adjusted AFLNet with the following configurations: ❶ Using the blackbox mode of AFLNet to make it available for protocol implementations with different programming languages besides C. ❷ Recording the maximum memory consumption on different state transitions during fuzzing process. We call AFLNet under the above configurations AFLNet* and used it as the baseline. AFLNet* will be released on our website [2].

**Evaluation Datasets.** We used 5 protocols (MQTT, FTP, DICOM, SMTP, and RTSP) which are commonly used in other protocol fuzzing works [1, 5, 8, 38] as our evaluation protocols. To select the programs for each protocol, we searched for implementations in five popular programming languages, including C, Java, JavaScript, Python, and Go, and finally selected 21 programs as the evaluation benchmark. Detailed information for these programs can be found in Appendix C or on our website [2]. The initial seeds used in fuzzing were obtained from ProFuzzBench [33]. Note that the same initial seeds were used for different implementations of the same protocol. The initial seeds and execution commands for each program will be released on our website [2].

**Evaluation Settings.** We ran all fuzzing experiments for 24 hours, to avoid bias caused by randomness [29]. We repeated each fuzzing campaign for 5 times and applied Mann-Whitney U test (*p-value*) [32] and Vargha-Delaney statistic ($\hat{A}_{12}$) [42] for statistic test.

**Experiment Environment** We conducted all experiments on machines with 80 cores of Intel(R) Xeon(R) Gold 6248 CPU @ 2.50GHz and 188 GB RAM. We ran each fuzzing experiment in the docker containers [15] with Ubuntu 20.04.3 LTS as the operating system.

**Table 2: Memory profiling results. For each attribute, the better mean value is highlighted in bold; the statistically significant ($p$-value < 0.05) value of $\hat{A}_{12}$ is marked with an asterisk**

| Protocol | Implementation | Avg mem (KB) | | | | Max mem (KB) | | | |
|---|---|---|---|---|---|---|---|---|---|
| | | AFLNET* | MEDUSA | | | AFLNET* | MEDUSA | | |
| | | mean | mean | ratio | $\hat{A}_{12}$ | mean | mean | ratio | $\hat{A}_{12}$ |
| MQTT | mosquitto (C) | 169.6 | **24054.4** | 141.83 | *1.00 | 529.6 | **73381.6** | 138.56 | *1.00 |
| | moquette (Java) | 568.0 | **2941.6** | 5.71 | *1.00 | 4175.2 | **62208.0** | 14.89 | *1.00 |
| | aedes (JavaScript) | 122.4 | **4830.4** | 39.46 | *1.00 | 11491.2 | **57091.2** | 4.96 | *1.00 |
| | hbmqtt (Python) | 94.4 | **22537.6** | 238.74 | *1.00 | 626.4 | **191334.4** | 305.45 | *1.00 |
| | hmq (Go) | 467.2 | **39271.2** | 84.05 | *1.00 | 35816.8 | **269080.0** | 7.51 | *1.00 |
| FTP | proftpd (C) | 3.2 | **733.6** | 229.2 | *1.00 | 186.4 | **3759.2** | 20.10 | *1.00 |
| | apache FtpServer (Java) | 1077.6 | **6372.0** | 5.91 | *1.00 | 6231.2 | **129020.0** | 20.70 | *1.00 |
| | ftp-srv (JavaScript) | 1367.2 | **25807.2** | 18.87 | *1.00 | 16963.2 | **120336.0** | 7.09 | *1.00 |
| | pyftpdlib (Python) | 96.8 | **2609.6** | 26.95 | *1.00 | 1448.8 | **22080.8** | 15.24 | *1.00 |
| | goftp (Go) | 269.6 | **4424.8** | 16.41 | *1.00 | 3934.4 | **21601.6** | 5.49 | *1.00 |
| DICOM | dcmtk (C) | 294.0 | **3788.8** | 12.86 | *1.00 | 738.4 | **7293.6** | 9.87 | *1.00 |
| | dcm4che (Java) | 3807.2 | **34453.6** | 9.05 | *1.00 | 5772.0 | **98204.8** | 18.59 | *1.00 |
| | pynetdicom (Python) | 472.8 | **13150.4** | 27.81 | *1.00 | 970.4 | **50437.6** | 51.97 | *1.00 |
| | go-netdicom (Go) | 704.8 | **77448.8** | 109.88 | *1.00 | 1461.6 | **157300.0** | 107.62 | *1.00 |
| SMTP | exim (C) | 77.6 | **1201.6** | 15.48 | *1.00 | 323.2 | **4877.6** | 15.09 | *1.00 |
| | Haraka (JavaScript) | 10.4 | **14920.8** | 1434.69 | *1.00 | 50.4 | **80295.2** | 1593.15 | *1.00 |
| | salmon (Python) | 4136.8 | **55048.0** | 13.3 | *1.00 | 70676.0 | **96088.0** | 1.35 | *1.00 |
| | go-guerrilla (Go) | 52.8 | **4512.8** | 85.46 | *1.00 | 136.0 | **9851.2** | 72.43 | *1.00 |
| RTSP | live555 (C) | 7.2 | **4652.0** | 646.11 | *1.00 | 40.0 | **7742.4** | 193.56 | *1.00 |
| | opencv-rtsp (Python) | 45135.2 | **46986.4** | 1.04 | 0.68 | 64420.8 | **64575.2** | 1.00 | *0.84 |
| | rtsp-simple-server (Go) | 259.2 | **13608.8** | 52.50 | *1.00 | 565.6 | **19984.8** | 35.33 | *1.00 |
| | **Average** | 2818.7 | 19207.3 | 153.11 | 0.98 | 10788.4 | 73644.9 | 125.7 | 0.99 |

## 4.2 Evaluation of Exploration (RQ1)

After the exploration phase, PPG profiles the memory consumption on different protocol state transitions of programs, in this experiment, we assess the PPG produced by the exploration phase to evaluate the MEDUSA's capability for profiling memory consumption. Specifically, we ran MEDUSA and AFLNET* on programs in the evaluation datasets, each experiment was run for 24 hours and repeated 5 times. Then we collected two type attributes of PPG: ❶ *Avg memory*. This attribute indicates the average memory consumption of all state transitions. ❷ *Max memory*. This attribute indicates the maximum memory consumption among all state transitions. These two attributes are used to assess fuzzer's ability for exploring resource consumption. For each attribute, we calculated the mean value of the results over all 5 runs and computed the ratio of MEDUSA's mean value over AFLNET*. We further calculated the *p-value* to measure the statistical significance of the results and $\hat{A}_{12}$ to measure the chance that MEDUSA can perform better than AFLNET* by randomly picking one result for comparison.

Table 2 shows the results. For memory consumption exploring, MEDUSA can discover up to 1593× max memory consumption than AFLNET*. Overall, MEDUSA discovers significantly bigger memory consumption than AFLNET* on almost all the programs with on average 153.11× avg memory consumption and 125.7× max memory consumption. The superiority of MEDUSA on avg memory consumption indicates that MEDUSA can outperform AFLNET* to discover more memory consumption for on average every state transition. From Table 2 we also observe that MEDUSA discover more memory consumption on 20 programs with the $\hat{A}_{12}$ value is 1.00 and *p-value* smaller than 0.05, which means that we have sufficient confidence to claim that MEDUSA has overwhelming superiority compared with AFLNET* for exploring memory consumption.

## 4.3 Evaluation of Verification (RQ2)

In this experiment, we evaluate the ability of MEDUSA in simulating DoS attack. Specifically, we used the PPGs generated from the exploration phase of experiments in Section 4.2. The experimental DoS environment is built upon docker containers. we used the breadth-first algorithm [45] to generate candidate traces from the initial state to the located stated transition. From the candidate traces we selected the trace which achieves the biggest cumulative memory consumption along the trace. We have tried different parameter values in the experiment and found that different parameters have little impact on the relative trend of the experimental results. Therefore, we adopted the following configuration for entire experiments and result presentation: ❶ Setting the memory limitation of the system to 4GB. ❷ Setting the monitoring duration to 60 seconds. ❸ Setting the attack time to 10 seconds after the test program starts up. ❹ Setting three level attack intensities: 1) *Low*. Sending 1 attack input per second. 2) *Medium*. Sending 10 attack inputs per second. 3) *High*. Sending 100 attack inputs per second.

**Memory Consumption.** Fig. 6 illustrates the memory consumption of the evaluated programs under simulated DoS attacks. From Fig. 6, we can observe that for the same attack intensity, attack inputs generated from MEDUSA can cause more memory consumption obviously than AFLNET* on almost all the programs. On hbmqtt, even the low attack intensity of MEDUSA can significantly outperform the high attack intensity of AFLNET*. Moreover, attack inputs generated from MEDUSA caused in total six programs (mosquitto, hbmqtt, hmq, apache FtpServer, ftp-srv, and go-netdicom) consume excessive memory (out of 4GB) and killed by the system. For AFLNET*, it only caused ftp-srv to consume excessive memory. We further analyze these cases in Section 4.4.

**Availability.** We also evaluated the availability of programs under simulated DoS. The details can be found in Appendix D.

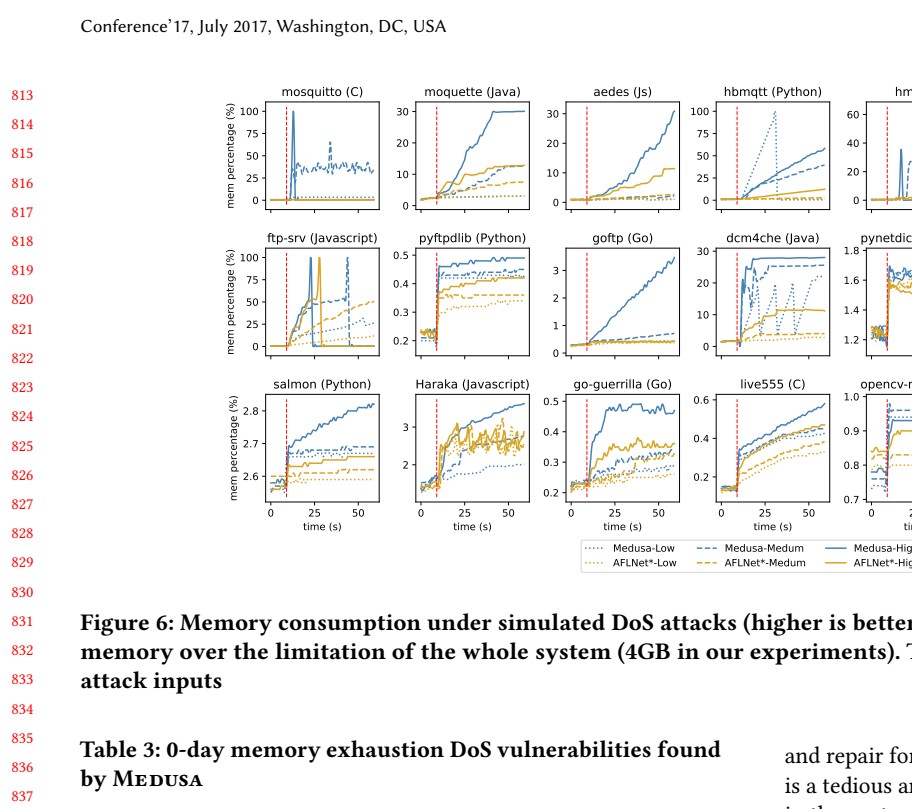

**Figure 6: Memory consumption under simulated DoS attacks (higher is better). Y-axis stands for the percentage of consumed memory over the limitation of the whole system (4GB in our experiments). The red vertical line is the time to start sending attack inputs**

**Table 3: 0-day memory exhaustion DoS vulnerabilities found by Medusa**

| Protocol | Program | Programming Language | Exhaustion Type |
|----------|---------|----------------------|-----------------|
| MQTT | mosquitto | C | System Memory Exhaustion |
| MQTT | hmq | Go | System Memory Exhaustion |
| MQTT | hbmqtt | Python | System Memory Exhaustion |
| FTP | apache Ftpserver | Java | Java Heap Memory Exhaustion |
| FTP | ftp-srv | JavaScript | JavaScript Heap Memory Exhaustion |
| DICOM | go-netdicom | Go | System Memory Exhaustion |

## 4.4 Vulnerabilities Detection (RQ3)

In Section 4.3, we discovered several cases that can cause the programs to exhaust their memory and be killed. To test the severity of these cases, we conducted simulated DoS attacks with different system memory limitations (16G, 32G, and 64G) and confirmed that these cases could indeed exhaust the system memory resources on all of the limitation settings. We further confirmed that these cases are memory exhaustion DoS vulnerabilities and still exist in the latest version of programs. Finally, we found six 0-day memory exhaustion DoS vulnerabilities, and the details are shown in Table 3.

Here we discuss the lessons we learn from the results of Table 3: ❶ Memory exhaustion DoS vulnerability is a serious threat existing in various protocol types (MQTT, FTP, and DICOM) and programming languages (C, Java, JavaScript, Python, and Go). ❷ We verified that the vulnerabilities do not exist in all implementation of a certain protocol, which emphasizes the need to focus on testing the specific protocol implementation instead of protocol specification. ❸ The exhaustion resource type is not limited to system memory, on apache Ftpserver and ftp-srv, the vulnerabilities exhaust the memory of language-internal components (Java VM [46] and JavaScript VM [47]). This demonstrates the damage of memory exhaustion DoS vulnerabilities as it may be much easier to exhaust the memory of language-internal components. ❹ The manual discovery

and repair for memory exhaustion DoS vulnerabilities in protocols is a tedious and prone to error process as these vulnerabilities exist in the protocol implementations rather than the specification. For example, although CVE-2017-7651 has been patched since 2018, the memory exhaustion DoS vulnerability still existed in Mosquitto for 5 years. it urges the demand for testing tools to help developers mitigate these vulnerabilities, Medusa is proposed for this.

## 5 RELATED WORK

Fuzzing techniques for protocol implementations can be classified into blackbox, whitebox, greybox methods according to information we obtain from the protocol to guide the fuzzing. Blackbox fuzzers from academia [7, 27, 28, 35] and industry [12, 19, 37, 41] treat protocol implementations as a blackbox and use either mutation-based or generation-based input generation technique to test the security of protocol implementations. They are preferred in industry since the fuzzing techniques do not pay attention to implementation details, which is scalable to test various protocols. Whitebox fuzzers [11] attempts to perform program analysis and guide the generation of input to execute different paths. Greybox fuzzers collect coverage or states to guide the input generation for testing the protocols. The way of identifying new code coverage and states ranges from human code annotations [6], invariants [17], response codes [38], to state variables [8].

## 6 CONCLUSION

In this paper, we propose Medusa, a dynamic analysis framework to unveil memory exhaustion DoS vulnerabilities in protocol implementations. Medusa utilizes a protocol property graph to guide exploring memory consumption in the explore phase and simulates DoS attacks to verify vulnerabilities in the verification phase. Our evaluation results demonstrate that Medusa can discover significantly more memory consumption in the exploration and verification phase compared to the baseline. Notably, Medusa found six 0-day vulnerabilities including one CVE ID.

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

# A THE METHODS OF EMPIRICAL STUDY

Here we discuss the methods we used to conduct the empirical study.

To identify the protocol resource exhaustion vulnerabilities from massive vulnerability information in the database, we first automatically filtered vulnerabilities with the following methods: ❶ Checking whether the type of vulnerabilities related to resource exhaustion. To automatically decide the type of vulnerabilities, we rely on the Common Weakness Enumeration (CWE) [14] metric. After investing the whole CWE list, we found several CWE types related to resource exhaustion vulnerability (CWE-400, CWE-401, CWE-404, CWE-770, CWE-789, CWE-1050, and CWE-1325). Vulnerabilities with at least one of the above CWE types are selected out. In addition, we checked whether the keyword `exhaust` exists in the description of vulnerabilities to complement some vulnerabilities which are related to resource exhaustion but without CWE type or assigned with wrong CWE type. ❷ Checking whether the vulnerabilities exist in the protocol implementation. For this, we checked whether the descriptions of vulnerabilities contain some keywords related to protocol. Specifically, we used `protocol`

and the name of several common-used protocol ("http", "mqtt", "ftp", "dicom", "smtp", "rtsp", "ssh", "tls", and "telnet") as keywords.

We combined and applied above two methods on CVE database with vulnerabilities from 2015 to 2022. Finally, we collected 205 vulnerabilities which related to resource exhaustion and exist in protocol implementations. Based on the collected vulnerabilities, we further conducted manual analyzing to identify the type of exhausted resource. The detail results and the scripts will be released on our website [2].

## B  IMPLEMENTATION DETAILS OF MEDUSA

State-aware fuzzer is implemented based on AFLNet [38]. To parse the protocol, it follows the same paradigm as AFLNet which uses embedded C functions to parse a test input into a sequence of request messages and extract protocol state from response messages. A new protocol can be easily supported by extending these functions. PPG uses graphviz [23] to record information and output it in file *ipsm.dot* which can be viewed using [24]. PPG records extra information related to state transitions (e.g. which message triggers the max resource consumption) and output in file *champion_josn* with JSON format.

As described in 3.2.1, Runtime Monitor is implemented base on Linux's */proc* Filesystem [10]. The main body of Runtime Monitor is a loop, when MEDUSA starts its monitor process, Runtime Monitor continuously accesses the *proc* Filesystem in the loop. To improve the efficiency of Runtime Monitor, we don't use the wrap library of */proc* Filesystem (e.g. psutil [20]), instead, we use file operators to directly access */proc* Filesystem and parse the resource consumption from the file content. Furthermore, as */proc* Filesystem is a Linux virtual file system [22] that its content updates in real-time every time reading it, we design the access process in the loop carefully to avoid frequently calling file *open* and *close* system calls. Specifically, we call file *open* at the initialization of Runtime Monitor, then in the loop before each time calling file *read*, we call file *lseek* function to move the file pointer to the head of the file. This implementation trick significantly improves the efficiency of Runtime Monitor.

## C  EVALUATION BENCHMARK

The programs used in evaluation are shown in Table 4.

## D  AVAILABILITY

Specifically, the availability is assessed by how many requests are processed by the program in a period of time. In this experiment, we additionally simulated a legal user sending requests (5 requests per second) to the program, we used throughput to represent the percentage of the requests that have been processed. 100% throughput means the program processes all the requests sent by the legal user which indicates a high availability. Fig. 7 shows the throughput of the programs under simulated DoS. Overall, the throughput has a negative correlation with memory consumption. The high-intensity simulated attacks with attack inputs generated from MEDUSA which cause the maximum memory increase can also cause the maximum decrease in throughput. This is because the attacks can lead to many

**Table 4: The real-world protocol programs used in the experiments.**

| Protocol | Implementation | Language | Version |
|---|---|---|---|
| MQTT | mosquitto | C | git commit ff97fbf |
| | moquette | Java | git commit 3e6043b |
| | aedes | Javascript | 0.46.3 |
| | hbmqtt | Python | git commit 07c4c70 |
| | hmq | Go | git commit b2e79c3 |
| FTP | Proftpd | c | git commit 0e68a6c |
| | apache FtpServer | Java | 1.2.0 |
| | ftp-srv | Javascript | git commit 18277e9 |
| | pyftpdlib | Python | git commit 2784660 |
| | goftp | Go | git commit f64f7c2 |
| DICOM | dcmtk | C | git commit c749632 |
| | dcm4che | Java | git commit 2f3165a |
| | pynetdicom | Python | git commit 5c2989e |
| | go-netdicom | Go | git commit 7caf23f |
| SMTP | exim | C | git commit a3d3e7e |
| | Haraka | Javascript | git commit 3198d18 |
| | salmon | Python | git commit a757003 |
| | go-guerrilla | Go | git commit aa54b3a |
| RTSP | live555 | C | git commit bbee4ed |
| | opencv-rtsp | Python | git commit 14d4d2c |
| | rtsp-simple-server | Go | git commit 8f48dfa |

memory-related operations, such as memory allocation, which reduces the overall performance of the program and increases the time delay for serving legal users, resulting in decreased throughput.

## E  ABLATION STUDY (RQ4)

To demonstrate the effectiveness of our optimization on Seed Selector and Message Mutator components, we conducted an ablation study by disabling optimization on these two components of MEDUSA and used the original strategies in AFLNET as comparisons. Specifically, we obtained three variants of MEDUSA and ran the additional fuzzing experiments as in Section 4.2 for them.

Fig. 8 shows the maximum memory consumption discovered by MEDUSA and it's three variants over all experiments. The results show that our optimization on both Seed Selector and Message Mutator contribute to MEDUSA for exploring memory consumption. This can be concluded from three observations on Fig. 8: ① The integral MEDUSA which enables the optimization on both Seed Selector and Message Mutator achieves the overall best result (the average median value for MEDUSA is 72197.14). ② The variant of MEDUSA (MEDUSA-DB) which disables the optimization on both Seed Selector and Message Mutator gets the overall worst result (the average median value for MEDUSA-DB is 64272.47). ③ For disabling the optimization on Seed Selector (MEDUSA-DS) and on Message Mutator (MEDUSA-DM) separately, although they have small differences from each other, they both achieve better results compared with MEDUSA-DB (the average median value for MEDUSA-DS and MEDUSA-DM is 69768.00 and 68072.19, which outperform MEDUSA-DB by 9% and 6% respectively).

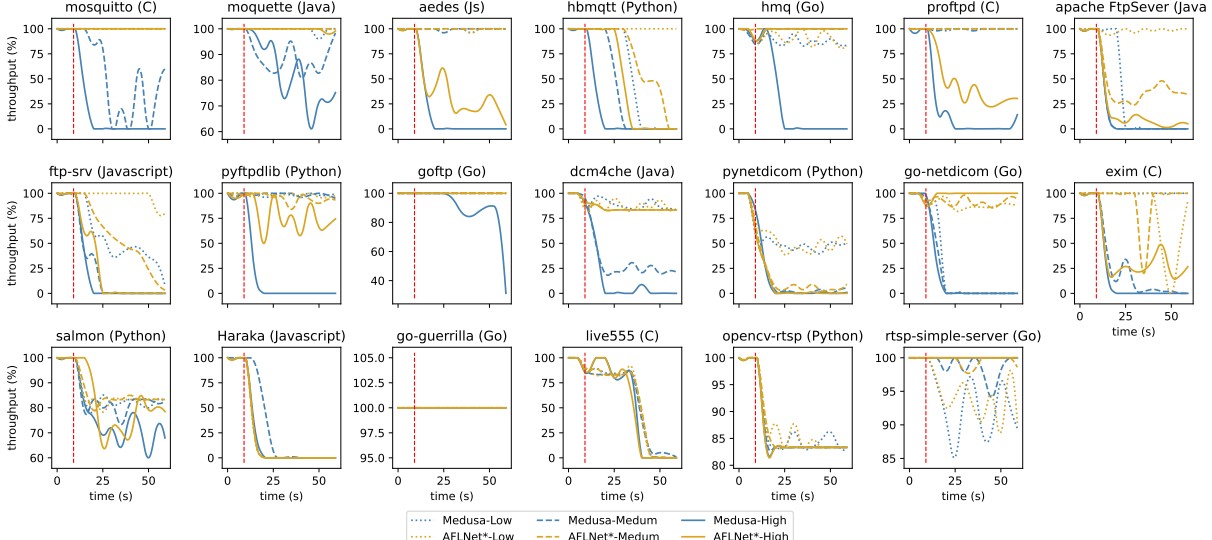

**Figure 7: Availability under simulated DoS attacks (lower is better). The red vertical line is the time to start sending attack inputs**

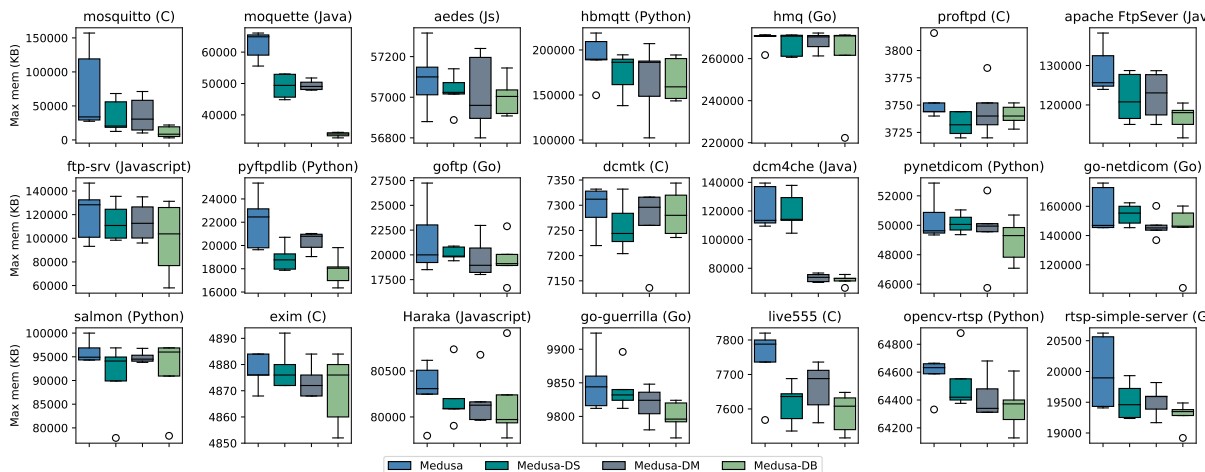

**Figure 8: Maximum memory consumption discovered after disabling resource-aware components (higher is better). DS = disabling Seed Selector; DM = disabling Message Mutator; DB = disabling both Seed Selector and Message Mutator**

