# OpenReview forum: "Medusa: Unveil Memory Exhaustion DoS Vulnerabilities in Protocol Implementations"
_ACM.org/TheWebConf/2024/Conference — TheWebConf24 Oral_

### Official Review · Reviewer_xcyh · 2023-10-25

**Novelty:** 7
**Technical Quality:** 6

**Review:**

The paper addresses the important and challenging problem of detecting memory exhaustion vulnerabilities in protocol implementations, especially in the context of preventing Denial of Service (DoS) attacks. It highlights the need for dynamic analysis techniques, as memory exhaustion vulnerabilities do not exhibit distinct code patterns, making them hard to identify through static analysis.

The paper effectively outlines the unique challenges in identifying such vulnerabilities and provides a clear problem statement:

1. Exploration of Memory Consumption: The paper emphasizes the need to explore memory consumption across various protocol states, which requires a well-planned strategy. This challenge is essential to understand how different states impact memory usage and vulnerability to DoS attacks.

2. Understanding the Risk of DoS Attacks: Excessive memory usage may not necessarily lead to DoS attacks due to various factors, including rate limitations imposed by the protocol. This issue highlights the complexity involved in assessing the feasibility of DoS attacks and the need for further verification.

3. Language-Agnostic Analysis: The paper acknowledges the presence of multiple protocol implementations in different programming languages and the importance of having language-agnostic analysis techniques to ensure generality.

The introduction of the "Medusa" dynamic analysis framework with its two phases (exploration and verification) is a significant contribution. Medusa utilizes the protocol property graph (PPG) to capture protocol states and related properties, enabling efficient exploration and validation of memory exhaustion DoS vulnerabilities. The paper also provides insights into evaluating Medusa's performance, including experiments conducted on various protocol implementations. The results are promising, showing that Medusa outperforms the baseline regarding memory consumption discovery and the ability to simulate DoS attacks. The discovery of 0-day vulnerabilities, with one assigned a unique CVE ID, demonstrates the practical significance of the approach. Overall, the paper effectively addresses a complex problem in the field of cybersecurity, presents a well-structured framework (Medusa) to tackle it, and demonstrates its practical utility through extensive experiments.

**Questions:**

The research is robust, and it presents its findings in a clear and easily comprehensible manner. The paper outlines a method for identifying protocol resource exhaustion vulnerabilities from a massive vulnerability database. The approach is broken down into two main steps:

1. Automatic Filtering of Vulnerabilities:

The authors use Common Weakness Enumeration (CWE) metrics to classify vulnerabilities and filter out those related to resource exhaustion. They identify specific CWE types (CWE-400, CWE-401, CWE-404, CWE-770, CWE-789, CWE-1050, and CWE-1325) associated with resource exhaustion vulnerabilities.
They also check the vulnerability descriptions for the presence of the keyword "exhaust." If the description contains this keyword, it is considered relevant to resource exhaustion, even if it lacks a specific CWE type or has an incorrect one.



2. Protocol-Related Vulnerability Filtering:

The authors check whether the vulnerabilities exist in the context of protocol implementations. To do this, they analyze the descriptions of vulnerabilities for keywords related to protocols. Specifically, they use terms like "protocol" and the names of commonly used protocols ("http," "mqtt," "ftp," "dicom," "smtp," "rtsp," "ssh," "tls," and "telnet") as keywords.
By combining these two filtering methods, the authors process a large dataset of vulnerabilities from the Common Vulnerabilities and Exposures (CVE) database, covering the years 2015 to 2022. As a result, they identified 205 vulnerabilities that are related to resource exhaustion and exist within protocol implementations. Subsequently, the paper indicates that the authors conducted manual analysis to further categorize the type of exhausted resource. It's mentioned that detailed results and scripts will be released at a later time. Overall, this methodology provides a systematic way to identify and filter vulnerabilities relevant to resource exhaustion in protocol implementations, helping to create a more targeted and manageable dataset for further analysis and research.

**Reviewer Confidence:**

4: The reviewer is certain that the evaluation is correct and very familiar with the relevant literature

**Scope:**

4: The work is relevant to the Web and to the track, and is of broad interest to the community

---

### Official Review · Reviewer_P648 · 2023-10-25

**Novelty:** 6
**Technical Quality:** 6

**Review:**

## Strength

+ The introduction of the protocol property graph (PPG) is novel and interesting
+ The generation of PPG is automated
+ The verification using PPG is interesting
+ The results are promising with six real-world vulnerabilities

## Weakness

- No evaluation of false positives and negatives
- No evaluation of code coverage

## Comments

This is an interesting paper with important results.  I like the idea of using PPG to verify and detect DoS vulnerabilities related to memory cons.  The results are important and demonstrate practical impacts.  That is why I recommend the paper to appear in theWebConf.   I do have some nitpicks for the paper though.

(1) The paper has no evaluation of false positives and negatives (Did I miss that?) It would be nice to see an evaluation of FPs and FNs, using (i) manually verifying the findings, and (ii) a curated dataset (e.g., the CVEs that you have studied).

(2) The paper does not have an evaluation of code coverage. Since it is a dynamic analysis framework, it would be nice to see how much code has been explored and whether there is unexplored code that may still be vulnerable.

To sum up, this is a good paper with significant impacts and I would like to see the paper to appear in theWebConf.

**Questions:**

1) What are your FPs and FNs?

2) What is the code coverage?

**Reviewer Confidence:**

3: The reviewer is confident but not certain that the evaluation is correct

**Scope:**

4: The work is relevant to the Web and to the track, and is of broad interest to the community

---

### Official Review · Reviewer_vD1o · 2023-11-20

**Novelty:** 3
**Technical Quality:** 3

**Review:**

## Summary
The authors propose MEDUSA, a dynamic analysis framework to detect memory exhaustion DoS vulnerabilities in protocol implementations. It generates a protocol property graph (PPG) including memory consumption information and uses it to simulate DoS attacks to verify the vulnerabilities. They evaluate their approach on 21 programs and the result shows that the proposed approach performs better than the existing techniques. Also, six zero-day vulnerabilities are detected, including one new CVE.

## Strengths:
1. Systematic method
2. Well written and organized
3. Performs better than existing approaches
4. Shares the code and dataset

## Weaknesses:
1. Weak novelty
2. Missing details in design
3. Lack of explanation of evaluation result

**Questions:**

### Weak novelty
The authors describe a framework to detect memory exhaustion bugs, but this is not considered as novel. The way to express the current information (e.g., memory consumption) of the program as a state (e.g., PPG) and state transition occurs according to the input of fuzzing is similar to the previous work by Bohme et al.* Also, the mutation operators used in this paper is basic operators (e.g., bit flip) with three additional operators (Replace, insert_begin, and Insert_end), which is not novel.
* (CCS 16) Coverage-based Greybox Fuzzing as Markov Chain


### Missing details in design
The explanation of constraints when mutation is insufficient. If the result of message mutation is problematic, there should be a way to avoid this. For example, there may be cases in which the packet becomes invalid after a bitflip is performed, making it unacceptable or state transition does not happen normally (staying in one state or moving around only in fixed multiple states).

In seed selection, according to 3.2.2, they design the score so that less selected seeds can receive higher scores by introducing SN(m), Select number.
There may be a way to use the seed more intensively when the seed causes more memory consumption, but the authors choose a way to explore more space by setting Select number (SN(m)) bigger than Memory consumption (TC(m)). I wonder if there are cases where promising seeds that may cause the bug are excluded by this mechanism. According to this logic, the currently discovered message causing DoS was discovered from a low-frequency seed. I am curious whether there are cases where a high-frequency seed produces a message causing DoS.
Additionally, this mechanism can be seen as an attempt to mitigate the local optimum, but I am curious whether the local optimum actually exists and whether there are cases where it actually escapes.


### Lack of explanation of evaluation result
For AFLNet*, the authors mention that they configure AFLNet* to record the maximum memory
consumption on different state transitions during the fuzzing process, it is unclear whether its mutation occurs in a direction that increases memory consumption.
Considering the result of Table 2, AFLNet* shows 3.2 and 7.2 for proftpd and live555, respectively, it is difficult to say that fuzzing was done with the goal of memory consumption. If so, more explanation is needed here.

The results show that AFLNet is not better than the proposed approach. However, it is difficult to understand why. Is this because AFLNet falls into the local optimum? On the other hand, the result of opencv-rtsp is almost equal to MEDUSA. If the authors add more explanation for this, it would be better.


### Nits
Duplicated sentences: “memory is the memory consumption incurred by the message; memory is the amount of memory consumption incurred by the message” in page 4

**Reviewer Confidence:**

3: The reviewer is confident but not certain that the evaluation is correct

**Scope:**

2: The connection to the Web is incidental, e.g., use of Web data or API

---

### Official Review · Reviewer_YRwM · 2023-11-22

**Novelty:** 5
**Technical Quality:** 6

**Review:**

## Summary
In this paper the authors designed and developed a fuzzing platform for identifying memory exhaustion vulnerabilities in network protocol implementations. They build a fuzzer named Medusa that is capable of measuring the memory overhead of various application states through building and exploring the protocol property graphs. They analyze 5 protocols and their respective implementations and report six zero-day vulnerabilities.

## Comments for the authors
The paper is well written and easy to follow and I enjoyed reading it. Overall I find the approach in the paper and the experiments to be reasonable and my feedback is positive. I have a few questions and comments that I will share below:

### Baseline for memory consumptions
In the third paragraph of page 2, the authors mention that they compared the memory consumption of applications under test with the baseline and identified 125.7x overall memory consumption. What is the baseline in this case? Is it the memory consumption when fuzzed with AFLNET? How does this compare to the memory consumption of the application at rest or under normal load?

### Takeaway from the empirical study of resource exhaustion vulnerabilities
In Section 2.2, the authors review the public CVEs for resource exhaustion vulnerabilities. They report that 64% of the reports belong to the memory exhaustion category. They then conclude that “memory is much more vulnerable to resource exhaustion DOS”. I’m not entirely sure if the data backs up their conclusion as there could be other factors (such as existing tools or ease of identifying certain categories of DOS vulnerabilities) that could explain why memory exhaustion vulnerabilities are identified and reported more than others. I want to hear the authors’ view on this and potentially rephrase the text to reflect this.

### Protocol specification
Under Methodology, the authors mention that in Step 4, MEDUSA builds the attack message sequences according to user-provided restrictions. What are some of these restrictions and are they required for proper operation of MEDUSA?
How does the complexity (i.e., size of the state machine) of target protocols affect the identified vulnerabilities? Are there examples of protocols where MEDUSA or other fuzzers have an easier or harder time reaching specific states?

### Related work
The details and comparisons with the prior work have been impacted in this paper most likely due to space limitations. That said, the authors should still contrast their work and position it with respect to other types of fuzzing (e.g., why is this method preferred to other blackbox or greybox fuzzers?)

**Questions:**

Repeating the questions from the review:

### Baseline for memory consumptions
In the third paragraph of page 2, the authors mention that they compared the memory consumption of applications under test with the baseline and identified 125.7x overall memory consumption. What is the baseline in this case? Is it the memory consumption when fuzzed with AFLNET? How does this compare to the memory consumption of the application at rest or under normal load?

### Protocol specification
Under Methodology, the authors mention that in Step 4, MEDUSA builds the attack message sequences according to user-provided restrictions. What are some of these restrictions and are they required for proper operation of MEDUSA?
How does the complexity (i.e., size of the state machine) of target protocols affect the identified vulnerabilities? Are there examples of protocols where MEDUSA or other fuzzers have an easier or harder time reaching specific states?

**Reviewer Confidence:**

3: The reviewer is confident but not certain that the evaluation is correct

**Scope:**

4: The work is relevant to the Web and to the track, and is of broad interest to the community

---

### Official Review · Reviewer_k9Fj · 2023-12-01

**Novelty:** 5
**Technical Quality:** 5

**Review:**

This paper addresses a crucial issue in web services security, focusing on the memory exhaustion vulnerabilities of protocol implementations. The proposed framework, Medusa, demonstrates promising results and contributes significantly to the existing body of knowledge on DoS attacks. In conclusion, the paper is well-structured, logically presented, and contributes significantly to the understanding and mitigation of memory exhaustion vulnerabilities in web service protocol implementations. However, there is still room for improvement in areas such as system safety analysis, program design, etc.

* Strengths
1. The topic of this paper is interesting and reasonable.
2. Medusa can satisfy more requirements than existing dynamic analysis techniques.
3. Extensive experiments have been conducted to validate the effectiveness and performance of the program.


* Weaknesses
1. There are many CVEs related to the content of this paper, why CVE-2017-7651 was chosen as an example, and what are the considerations? The above issues are not well articulated in the paper.
2. The performance of some parts of the program is improved, but it is not analyzed and only experimental results are given briefly.
3. Possible follow-up work based on this work is not given.
4. The authors did not provide a theoretical analysis of the system, such as performance, system security, etc.
5. No corresponding preventive measures and recommendations were given.

**Questions:**

1.	The solution in this article has more features to meet more requirements. Are there any the resulting disadvantages?
2.	Are there potential security issues with this system?

**Reviewer Confidence:**

3: The reviewer is confident but not certain that the evaluation is correct

**Scope:**

3: The work is somewhat relevant to the Web and to the track, and is of narrow interest to a sub-community

---

### Decision · Program_Chairs · 2024-01-22

**Decision:**

Accept (Oral)

**Comment:**

# Summary

 This paper addresses the problem of how to identify Memory Exhaustion DOS attacks. The approach uses dynamic analysis (fuzzing), and makes several modifications to real-world fuzzing to accomplish this goal. The presented system was evaluated on several applications and found several real-world vulnerabilities, including a previously unknown vulnerability.

 # Strengths

 + The topic of this paper is interesting and a reasonable fit for TheWebConf.
 + The paper has extensive experiments (which will be extended in the final version with a comparison with AFLNET).
 + The paper is well written and organized.
 + The code and data will be open-sourced and shared with the community.

 # Weaknesses

 - No discussion of False Positives / False Negatives (these will be added).

 # Recommendation

 Overall, the reviewers are in agreement that this is a strong paper with strong relevance to TheWebConf. We expect the authors to make all necessary changes to the paper as they agreed to do in the interactive discussion phase.

 ---